# Acute Pain Service in Hungarian hospitals

Orsolya Lovasi[1]*, Judit Lám[2], Réka Schutzmann[1], Péter Gaál[2]

1 School of PhD Studies, Semmelweis University, Budapest, Hungary, 2 Semmelweis University Health Services Management Training Centre, Budapest, Hungary

* lovasi.orsolya@gmail.com

## Abstract

### Background

Surgical procedures play an increasing role among health technologies to treat diseases. Pain often accompanies such diseases, both as a result of their pathology, but also as the side-effect of the intervention itself, and it is not only a burdensome subjective feeling, but adversely affects the recovery process, can induce complications and increases treatment costs. Acute Pain Service Teams are becoming increasingly widespread in hospitals to address post-operative pain, yet we have so far no data on how many hospitals have actually adopted this technology in Hungary.

### Objectives

The main objectives of our study were to assess the prevalence of Acute Pain Service Teams, map their structure and operation, as well as to understand the barriers and conducive factors of their establishment in Hungarian hospitals.

### Methods

We carried out a survey among the 72 hospitals with surgical departments. The questionnaire was filled in by 52 providers, which gave us a response rate of 72.2%.

### Results

Our results show, that only two of the responding hospitals have Acute Pain Service Teams albeit their structure and operation are in line with the literature. In the 50 hospitals without such teams, financing difficulties and human resources shortages are mentioned to be the most important obstacles of their establishment, but the lack of initiative and interest on the part of the specialities concerned are also an important barrier.

### Conclusions

Lagging behind the more affluent EU member states, but similarly to other Central and Eastern European countries, Acute Pain Service has been hardly adopted by Hungarian hospitals. Hungarian health professionals know the technology and would support its wider introduction, if the technical feasibility barriers could be overcome. Health policy should play

**Data Availability Statement:** All relevant data are within the manuscript and its Supporting Information files.

**Funding:** The authors received no specific funding for this work.

**Competing interests:** The authors have declared that no competing interests exist.

a more active role to facilitate change in this area, the investment in which promises a substantial return in terms of health gains and cost savings.

## Introduction

There is an extensive literature on the importance of pain and pain management in patient care, which stems from its biological, psychological and economic consequences [1, 2]. First, pain is a burdensome subjective feeling, defined as "an unpleasant sensory and emotional experience associated with, or resembling that associated with, actual or potential tissue damage," by the International Association for the Study of Pain [3]. Second, uncontrolled pain has adverse impact on the effectiveness and efficiency of patient care, through the hyperactivity of the sympathetic nervous system, stress reaction, changes in the endocrine and nervous system, induced sleep deprivation, anxiety and depression, reduced mobility, increased risk of pneumonia, and myocardial ischemia [1, 4–7]. In addition to these pathologies, uncontrolled postoperative pain also increases the costs of patient care through longer hospital stay and the treatment of complications [7]. Third, if not appropriately managed, pain can persist over longer periods of time, prolonging its complications [1, 8]. Chronic pain, i.e. pain that lasts for more than three months [2], can result in increased nociceptive sensitivity through multiple signaling pathways. This peripheral sensitization leads to allodynia and hyperalgesia, which in turn result in central sensitization by decreasing inhibitory modulation in the dorsal horn and the brainstem [9].

Although surgical interventions are an option to treat the pain caused by a disease, they are also important factors in generating pain as the side-effect of the invasive procedure itself [10]. Coluzzi et al. [11] has estimated that the prevalence of severe pain on the first day after the operation is 39%, and after certain surgical procedures more than 75% of patients experience moderate or serious pain, while according to the estimation of Petti [7], the frequency of prolonged postoperative pain can be between 5% to 85%, depending on the type of the operation (e.g. limb amputation, breast surgery, hysterectomy, total hip replacement) [2]. The significance of postoperative pain is increasing, along with the volume of operations performed, whose number worldwide was estimated to exceed 300 million per annum in 2012 [12, 13].

In Hungary the research on postoperative pain is scarce. In one of these rare studies Lovasi et al. [14] assessed the severity of postoperative pain among 168 surgical patients of three Hungarian hospitals, on the basis of self-rating on a scale of 0–10, where 0 represented the lack of pain and 10 represented unbearable pain. Right after the operation, 54% of the respondents experienced level 5 or stronger pain, while the corresponding figures 12 hours after the operation and 24 hours after the operation were 55% and 42%, respectively.

In summary, the frequency and consequences of postoperative pain justifies to pay more attention to it, and there is evidence that effective postoperative pain treatment can minimise the complications and costs associated with uncontrolled pain and increase patient satisfaction at the same time [4, 6, 8].

Effective postoperative pain management includes regular pain assessment, pain relief protocols, and pain therapy administered at the right time [15]. Emerging evidence suggests that pain treatment can be more effectively managed by a multidisciplinary team of health professionals, referred to as Acute Pain Service (APS), rather than individual nurses or doctors, given the complexity of the technology. There is ample evidence, for instance, that APS managed pain therapy increases the quality of postoperative pain treatment, decreases the pain

intensity in patients, improves the health status of patients, decreases the prevalence of side effects (such as nausea and vomiting), reduces the length of stay, and improves overall patient satisfaction and patient safety [16–19]. Further, APS plays an important role to improve the efficient use of scarce health care resources, decrease costs by preventing side-effects and reducing length of stay [6]. The cooperation of specialists, nurses and other health professionals in all the components of effective pain management is the critical success factor of APS, including pain assessment, pain therapy, team organization and education [20]. The deployment of APS enhances the effectiveness of postoperative pain management through multiple intervention points, since in addition to the supporting of evidence-based clinical practice, the promotion of the wide-scale application of pain relief technologies and the coordination of clinical activities, they manage the information of patients, the education of the staff and the indicator-based systemic evaluation of the whole process, which improve the transparency and accountability of health care [21, 22].

So far, three types of APS have been described in the literature, distinguished by the lead health professional of the team. The earliest model was introduced by Ready, whereas the team is headed by anesthesiologists [23]. Taking into account the high cost of specialists, especially anesthesiologists, a second version was developed in Sweden, according to which the actual management of the team is carried out by trained nurses, who are supervised by anesthesiologists [20, 24–27]. Finally, Borracci et al. [28] proposed a third option, whereas the APS is run by young doctors in anesthesiology residency training.

Nevertheless, postoperative pain management is still a challenge for health professionals, health service providers and health systems alike [19]. For instance, according to a study of 2,250 German hospital patients, 50% of them considered the postoperative pain therapy inadequate [29], and Hungary is no exception. We have hardly any reliable information on the practice of postoperative pain management, and the adoption of APS by hospitals. We have found no study about what percentage of Hungarian hospitals manage postoperative pain by multidisciplinary teams, what the composition of these teams are, and what rules govern their operation, despite the convincing body of evidence, summarised above, on their beneficial impact and widespread use internationally.

The aim of our study, therefore, was to assess whether or not APS were used in postoperative care at all in Hungary, how prevalent the use of multidisciplinary teams was, and to map their structure and operation. Based on our own experiences, we hypothesised that APS was not the dominant form of postoperative pain management in Hungarian hospitals, so we also wanted to explore and understand the barriers to their establishment among those providers, which had no APS, yet. For this purpose, we implemented an explorative survey on APS among Hungarian hospitals, which had surgical inpatient care.

## Materials and methods

### Compiling the questionnaire

To address the research questions, we carried out a survey among all the 72 Hungarian hospitals, which had surgical inpatient care. We obtained the list of inpatient care providers from the National Public Health Center, the authority responsible for issuing operational licences to health service providers in Hungary. We included in the sample each and every hospital, which had at least one valid licence to operate inpatient care in surgery, orthopedics and/or traumatology, regardless of their owners, or financing (public or private), and provide actual surgical services on the basis of the issued licence.

The survey process was pilot tested before data collection began. The pilot questionnaire was developed on the basis of a review of the English language international literature. We

identified published questionnaires [1, 21, 22, 30, 31], selected the relevant questions and translated and adapted these to the Hungarian context. The pilot questionnaire was tested by a group of health professionals, including three anesthesiologists, two general surgeons, an orthopedic surgeon, two anesthesiology assistants, and a highly qualified nurse, who were requested to fill it in and provide feedback on any format and language issues that might affect the accuracy of the answers obtained. In addition, we also asked for the opinion of the experts of two professional organizations, the Hungarian Pain Society and the Hungarian Society of Anaesthesiology and Intensive Therapy. The questionnaire was finalized on the basis of the feedback we received from these sources.

The final questionnaire, which mainly contained closed questions with the option to complement the answers with open-ended comments, covered four main topics: the general characteristics of the institution, the structure and operation of APS, data collection regarding pain management, and the reasons of not having an APS, the perceived barriers to its establishment (Table 1). The adapted questionnaire had 67 questions. In the frame of this paper, we present the analysis of questionnaire items regarding the presence and function of APS.

## Data collection

The survey was implemented from the beginning of December 2019 till the end of February 2020, just before the outbreak of the COVID-19 epidemic in Hungary. The questionnaire was distributed electronically among the identified hospitals, together with brief instructions on its completion. Each institution was requested to fill in one questionnaire per specialty, which meant that we expected to receive at least two and maximum four questionnaires from each hospital. This allowed us to obtain data from more than one person, and made the analysis of different perspectives possible. Nevertheless, if a hospital had APS, the related questions had to be answered only by one of the anesthesiologists. This means that we obtained data on the APS at the level of the hospital, while all other questions (related to the barriers of their introduction) could be answered by more than one specialist, and maximum four, if all three surgical specialties were represented in a hospital.

The ethical approval of the research was obtained from the Scientific and Research Committee of the Medical Research Council (approval number: 51962-2/2019/EKU) [32]. The

**Table 1. The main topics of the questionnaire.**

| Main topics | Questions |
|---|---|
| 1. Hospital characteristics | • Geographical location of the hospital by region?<br>• Respondents' position, speciality?<br>• Owner of the hospital? |
| 2. Availability, structure and operation of APS | • Is there an APS team in the hospital, for how long?<br>• Who are the members of the team, do they work full time or part time in the APS?<br>• Who are treated by the APS (only surgical patients, or others as well)?<br>• Out-of-hours operation?<br>• Financing of the APS? |
| 3. APS data collection | • What are collected and how (electronic or paper based data collection)?<br>• How are the collected data processed? |
| 4. Practice of pain management, barriers to set up APS | • If there is no APS in the hospital, who is responsible for pain management?<br>• If there is no established APS in the hospital, why is that?<br>• What are the barriers of their establishment? |

questionnaires were sent to hospital owners, and consent of the participants of the study to use the obtained data was granted by them by returning the filled in questionnaire. The participants did not get any compensation for completing the questionnaire.

## Data analysis

Given that there have been no research, which addressed the prevalence, structure and operation of APS previously in Hungary, our study was exploratory relying mainly on descriptive statistical analysis. The low number of hospitals, which has already adopted the APS technology, made the use of more elaborate statistics impossible. The collected questionnaires were analysed in a descriptive manner, with the answers expressed as frequencies and percentages. The statistical analysis was carried out by the Statistical Package for the Social Sciences (SPSS) version 24.0 (IBM Corp., Armonk, NY, USA).

All together 72 hospitals were identified on the basis of the data of the National Public Health Center, out of which 52 responded to our request providing us a 72.2% response rate. In total, 20 hospitals did not return filled questionnaires, 12 government hospitals, two university clinics, one specialized hospital and five private for-profit hospitals (Table 2). Given that most hospitals had more than one specialty covered, we got back almost three times as many, a total of 145 questionnaires out of which 135 could have been analysed. We excluded the questionnaires, where all, or most of the questions of section 4 were left unanswered by the respondent. However, not all of the remaining 135 questionnaires were 100% complete, so the actual number of responses, indicated later by "N", can be lower for certain items due to missing answers. The distribution of hospitals, response rates and the number of questionnaires is shown in Table 2, according to the owner, or authority responsible for the maintenance of the institution, while Table 3 summarises the department and position of those health professionals, who actually filled in the analysed questionnaires.

## Results

### The prevalence and operation of Acute Pain Services in Hungary

Only two hospitals out of the 52 reported operating APS. Both are governmental hospitals under the supervision of the Ministry of Human Capacities, they are located in the countryside (neither of them in the capital, Budapest): one in the Northern Great Plain (Team 1), the other in the Western Transdanubia (Team 2) region of Hungary. Both APS are relatively new, they were established less than five years before the survey. As far as the subjective evaluation of the operation of APS is concerned, Team 1 is considered "flourishing", while Team 2 is "struggling". The size and professional composition of the two teams are different. Team 1 consists of

**Table 2. Description of the sample of institutions and response rates by owners.**

| Hospital owner | Number of identified hospitals | Number of responding hospitals | Response rate | Number of questionnaires analysed |
|---|---|---|---|---|
| Government hospitals under the Ministry of Human Capacities | 57 | 45 | 78.9% | 111 |
| Clinical centres of universities under the Ministry of Innovation and Technology | 5 | 3 | 60.0% | 13 |
| Special hospitals under other ministries | 2 | 1 | 50.0% | 4 |
| Private, Church-owned hospitals | 1 | 1 | 100.0% | 2 |
| Private for-profit hospitals | 7 | 2 | 28.6% | 5 |
| **Total** | **72** | **52** | **72.2%** | **135** |

**Table 3. Distribution of the respondent health professionals by position and place of work.**

| Position/Department | Anesth.† | Surgery | Trauma.† | Orthop.† | Total |
|---|---|---|---|---|---|
| Head of department | 38 | 16 | 16 | 10 | **60.2%** |
| Head of unit | 2 | 2 | 0 | 0 | **3.0%** |
| Specialist | 7 | 11 | 7 | 2 | **20.3%** |
| Head nurse | 2 | 5 | 4 | 1 | **9.0%** |
| Doctors in residency training | 1 | 2 | 2 | 3 | **6.0%** |
| Nurse | 1 | 0 | 1 | 0 | **1.5%** |
| **Total** | **38.3%** | **27.1%** | **22.6%** | **12.0%** | **100.0% (N = 133)*** |

Note: *Out of the 135 respondents, 2 did not answer the question on their position, so they could not have been included in this analysis.

†Anesth.; Anesthesiology, Trauma.; Traumatology, Orthop.; Orthopedics.

three main professionals, who are exclusively anesthesiologists, while Team 2 is based on allied health professionals (including nurses and anesthesiology assistants), whose work are supervised by an anesthesiologist. Other health professionals, such as surgeons, neurologists, pharmacists, nurse practitioners, physiotherapists, specialised pain nurses, or psychologists are not included in these teams, but psychiatrists are available for consultation. There are no dedicated team members in either APS, all of them serve in the team parallel to their other work duties.

Both teams carry out their work in a matrix structure, based on a written agreement with the relevant departments to treat their patients, but without any financial compensation. Team 2 care for all the patients, who underwent an operation, while Team 1 only for special cases with either epidural analgesia cannula (EDA), intravenous patient-controlled analgesia (IV-PCA), or easy pump. Beside postoperative pain management, both teams serve non-surgical patients as well, but they do not exceed 20% of their workload. Out-of-hour services are only provided by Team 1. In the case of Team 2, patients are attended by the regular out-of-hours service of the hospital, with an anesthesiologist on duty, who can be reached via mobile phone. The operation of both teams are guided by written treatment protocols.

## Data collection, administration and documentation in Acute Pain Services

The administration of patient care with detailed data collection and documentation is part of the operation of both teams. The documentation is exclusively paper based in Team 2, while Team 1 complement the paper based documentation with electronic recording and processing of data with dedicated computer software. The content of the documentation is similar, patient visits are recorded in both institutions. The regularly generated and recorded data include, among others the name, age and gender of the patient, the type of operation, the type of anesthesia, the health status of the patient according to the American Society of Anesthesiologists Physicial Status Classification System (ASA), pain assessment scores, the measurement of the efficiency of cannula, but adverse events, potential complications (e.g. cannula displacement), and unsuccessful interventions are also recorded. Team 1 uses a dedicated APS observation sheet, on which the main patient identification data (name, ward, social insurance identification number), the name of the person, who inserted the cannula, the date, time and place of the insertion, the administered medicine, the vital parameters of the patients (tension, pulse) and the pain score (on a numerical rating scale, 0–10) are registered. Team 1 shares the experiences with other hospitals, Team 2 with its own colleagues, but neither of them share their experiences with their fellow surgical wards and the hospital management.

The main characteristics of the two APS teams, compared to each other, are summarised in Table 4.

**Table 4. The main characteristics of operational Acute Pain Services (APS) in two Hungarian hospitals.**

| Characteristics | Team 1 (based on anesthesiologists) | Team 2 (based on allied health professionals) |
|---|---|---|
| Structure and composition of teams | Based on medical doctors | Based on allied health professionals supervised by an anesthesiologist |
| Out-of-hours service | Yes | No |
| Dedicated personnel | No | No |
| Representation of other professions | No | No |
| Regular pain assessment appropriately documented | Yes | Yes |
| Written pain treatment protocols | Yes | Yes |
| Dedicated financing for the team | No | No |
| Documentation technology | Paper-based, processed by dedicated computer software | Paper-based |
| Content of the documentation | • Patient identification data<br>• Type of operation<br>• ASA* classification level<br>• Pain scores†<br>• Adverse events<br>• Dedicated APS sheet | • Patient identification data<br>• Type of operation<br>• ASA* classification level<br>• Pain scores†<br>• Adverse events |

Note: *ASA = American Society of Anesthesiologists Physical Status Classification System,

† Pain scores as measured on a numerical rating scale of 0–10.

## The need for and barriers to the establishment of Acute Pain Services in Hungary

Among the respondent departments from hospitals without established APS (N = 115), the overwhelming majority (82.6%, n = 95) think that such multidisciplinary teams would be necessary, while less than one fifth (17.4%, n = 20) think that they are not needed. Among the respondents, who think that APS is necessary, 63.2% (n = 60) were heads of department, 18.9% (n = 18) specialists, 6.3% (n = 6) doctors in residency training, 6.3% (n = 6) headnurses, 2.1% (n = 2) ward nurses, 2.1% (n = 2) other position, and one respondent with missing position. Regarding their distribution according to departments, 47.4% (n = 45) work at an anesthesiology and intensive care unit, 23.2% (n = 22) at a surgical department, 17.9% (n = 17) at a traumatology, and 11.6% (n = 11) at an orthopedics department.

In terms of the barriers to establish APS, respondents could choose more than one out of 7 predefined answers (availability of equipment, availability of medicines, lack of financing, lack of initiative and motivation, lack of support from the management of the hospital, lack of interest on the part of colleagues, insufficient cooperation between different specialisations, lack of human resources), but they were also offered the option to mention other reasons (Fig 1). Respondents identified an average of 2.7 (SD = 1.25; N = 112) barriers. Lack of human resources has been mentioned most frequently (75%, n = 84), followed by the lack of financing in the second (43.8%, n = 49) and the lack of initiative (38.4%, n = 43) in the third place. Over one third of the respondents named the lack of interest (35.7%, n = 40) and the lack of cooperation among different specialists (35.7%, n = 40) as an important feasibility barrier, while less than 15% selected the lack of support from the management (14.3%, n = 16), and the lack of equipment and medicines (14.3%, n = 16). Other barriers, including the lack of knowledge, the lack of demand, the lack of national regulation, the lack of officially acknowledged training in pain therapy, the small size of the hospital and few patients, were mentioned by 11.6% (n = 13) of the respondents.

Further, we asked the respondents (N = 128) whether they would be willing to participate in the establishment of APS in their institutions. Close to two thirds answered yes (62.5%,

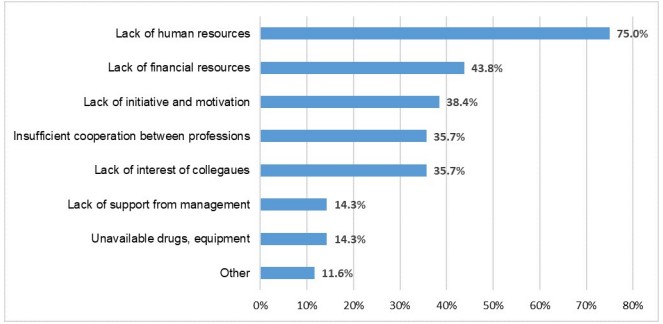

**Fig 1. Barriers to the establishment of Acute Pain Services (APS) in Hungarian hospitals, based on the opinion of respondent health professionals (N = 112).**

n = 80), 10.9% (n = 14) of the respondents answered no, and 26.6% (n = 34) provided no answer.

We also wanted to explore the opinion of respondents on whether or not it was the responsibility of the anesthesiologists to initiate the establishment of APS, and why (N = 119). About 80% of respondents said the anesthesiologist should take responsibility for the operation of APS, (80.7%, n = 96) and 19.3% (n = 23) said the opposite. The reasons can be grouped into three main categories. First, most of the respondents argued that anesthesiologists were best qualified for this task (they are the most competent, most experienced, most knowledgeable, best placed to monitor the perioperative period, and have greatest insight into pain management technologies). Second, some respondents emphasized the need for cooperation between the anesthesiologist and the operating specialists. While they think that the primary responsibility for pain relief lies with the anesthesiologists, especially during the early postoperative phase, it is not possible to assign an anesthesiologist to each patient, so representatives of surgical specialisations have an important role to play. Third, some respondents have taken the position that everybody should be responsible for their own patients, and there is no need for cooperation with other professions ("own patient-own department" category).

## Who is responsible for pain management in the absence of Acute Pain Service?

The sample is distinctly heterogeneous in that, how postoperative pain management is organised in the absence of APS (N = 128). It is exclusively carried out by the operating department in the case of 33.6% (n = 43) of the respondents, while 10.9% (n = 14) indicated the exclusive responsibility of the personnel of intensive therapy departments, or pain clinics. Nevertheless, most respondents described some form of cooperation among various organizational units and professions (55.5%, n = 71). The most frequently mentioned option was the cooperation between the department, where the patient stayed and the anesthesiologists, but other combinations also appeared among the answers, for instance the staff of the pain clinic and surgical specialists of the operating department.

## Discussion

The results of our study suggest that APS is not widely used in postoperative pain management in Hungarian hospitals. From a cross-country comparative perspective, we can say that the wealthier European countries, with high level of public spending on health started the introduction of APS decades ago, and the technology has been gaining ground at a fast pace ever

since [11, 21, 33, 34]. The finding that only two (3.9%) out of the 52 hospitals participating in our study has APS implies that Hungary is just beginning to join this trend, with a 25-year lag.

In one of our earlier research, we reviewed the literature to identify, when other countries had begun to adopt this organisational innovation, that is the early 1990s, and most of these countries have experienced further development since then [14]. For instance, Erlewen at al. found that 81% of hospitals in Germany operated APS in 2015 [33]. By 2012, 58.5% of Italian hospitals adopted this technology [11]. In the Netherlands, hospitals are obliged to operate APS, which is part of the Dutch Hospital Patient Safety Program of the government, launched in 2008, among others to decrease avoidable suffering from pain by early detection and adequate pain therapy, and APS coverage has reached 90% of inpatient care providers [18]. In Denmark, the proportion of APS among university hospitals was 52–71%, while in regional hospitals the corresponding figure was 8–40% in 2009 [34]. According to a Portuguese survey, APS coverage was 47.7% in 2012 [24]. In the neighbouring Austria, 39.2% of hospitals providing surgery had APS in 2011 [35].

As far as post-socialist Central and Eastern European countries are concerned, survey data on the prevalence of APS are scarce, but there is evidence that they know about the concept, the need for multidisciplinary pain teams is recognized, and the introduction has already begun, albeit the development process is still in its infancy. For instance, a Czech study investigated the association between chronic postoperative pain and the availability of APS, and the research found lower pain intensity among hospitals with APS. This implies that Czech hospitals have already started to adopt the technology [36] A Serbian survey found that patients were underinformed about postoperative pain relief, which, according to the researchers, could be improved with the introduction of APS [37]. In a small scale Hungarian study of three hospitals, Lovasi et al. [38, 39] found that although none of the studied providers operated multidisciplinary pain teams, 60% of the doctors would have supported their introduction.

In terms of the organizational structure and operation of APS, our findings suggest that they by and large correspond to the models described in the literature. One of the identified teams was found to consist only of doctors, in particular anesthesiologists, while the other was based on allied health professionals, supervised by an anesthesiologist. The nurse-based APS is the more cost-effective option, as specialized nurses and nurse practitioners can be well deployed to perform most APS tasks and cover out-of-hours operations [20, 33, 40]. In contrast, doctor based teams are better at addressing more complicated cases and scenarios, such as chronic postoperative pain, addiction, palliative care and consultation with other specialists [33].

While the backbone of APS in other countries also consists of doctors and nurses, marked difference is that they typically involve other health professionals, such as pharmacists, physiotherapists, psychologists and neurologists [21, 26, 41]. The involvement of surgeons is an interesting question, as it is not only Hungarian teams, which forego their contribution. There are no surgeons among the team members even in The Netherlands, where APS penetration is almost complete. Increasing their involvement might be an important intervention point to enhance the effectiveness of APS through better cooperation [18]. This could also be an important facilitating factor in Hungary, where the operating APS teams do not share the information with neither the surgical specialists, nor with the hospital management. The sharing of favourable experiences would contribute to the acknowledgement of this organizational solution among these important stakeholder groups in the hospitals, which in turn would support its faster adoption.

The functioning of the identified Hungarian APS teams by and large also corresponds to the models described in the international literature, with some notable differences. Stamer et al. [42] described 5 key characteristics of high quality APS: (1) dedicated personnel, (2) the continuity of care during out-of-hours periods, (3) the application of written pain treatment

protocols, (4) regular pain assessment, and (5) administration and documentation of the care process [33, 42]. Among these criteria, the Hungarian teams fall short of having dedicated health professionals, working exclusively in the APS, and of providing service coverage during nights and weekends. On the other hand, other countries, with more developed APS, also report difficulties in these areas. For instance, in The Netherlands only 58% of hospitals have pain teams with at least one dedicated health worker [18]. According to a British study, the APS of only 15% of hospitals provided services during night duty, while the corresponding figure was 29% regarding weekend duty [31]. In many cases, 24h coverage was provided via mobile phone by anesthesiologists or nurse practitioners [1, 18, 40].

The administration and documentation of the operation of APS are similar to other countries and corresponds the criteria described by Duncan at al. [1]. For instance, similar activities with similar data are recorded by the identified Hungarian teams as in Germany [33].

## Barriers to the uptake of the technology: Human resources and education, motivation and financing

Given the very high number and proportion of Hungarian hospitals without APS, identified by our study, the section of the research, which addressed the barriers of the introduction of APS perceived by the respondent health professionals, brings an even greater added value into the research of pain management practices in Hungary. According to the relevant literature, the main obstacles include the lack of financing, the shortage of qualified human resources, overloaded staff, lack of knowledge and the insufficient interest and motivation of health service providers [30, 33]. Most of the worktime of anesthesiologists are consumed by assisting operations, and little time remains for postoperative pain visits in wards [30]. Close to half of the respondents of an Italian study considered the lack of time and organizational difficulties as the most important barrier of the implementation of optimal postoperative pain management, but ignorance also plays a role [11]. For instance, a still widely held belief among specialists that postoperative pain relief is needed only for a few days after the surgery, which does not justify the costs and efforts devoted to the establishment and operation of APS or the application of advanced pain relief technologies, such as patient controlled anesthesia. Others fear of the potential side effects and complications of such technologies [11].

In Hungary, the shortage of qualified human resources is a key barrier to a wider adoption of APS by hospitals. Hungary is among the European countries, which is most hit by the emigration of health professionals, especially of specialists, such as anesthesiologists, for whom there is an increasing demand in more affluent member states of the European Union [43–46]. One strategy to address these shortages is training and education. Losses can be compensated either by training more health professionals, doctors and nurses alike, or by training highly qualified allied health professionals, such as advanced nurse practitioners to take over certain tasks from doctors (task shifting). A specialized APS nurse practitioner would be able to assist certain pain therapies, such as nerve block anesthesia, by preparing the necessary equipment, administering the necessary sedatives, positioning the patient, assisting the placement of the block, the anesthesia and the observation of patients after the intervention [47]. While the training of advanced nurse practitioners is already legally recognized and implemented in Hungary, the necessary competencies have not yet been legally conferred on them due to the strong opposition from the medical profession. Without the necessary official authorization, no such advanced nurse practitioners can be actually deployed in patient care.

Education and training is also the cornerstone of team members and cooperating health professionals possessing the necessary knowledge and information to ensure the effective operation of the APS. Some respondents of our survey mentioned the lack of an officially

acknowledged special pain management license as a possible barrier to the introduction of APS in hospitals. Indeed, the depth of knowledge justifies a special training programme and examination in this area. Some authors propose the modification and expansion of the residency training of anesthesiologists [27, 47]. These are not only related to medical knowledge, i.e. to be the expert of regional anesthesia, but also to educational skills to be able to train medical students, residents and other colleagues, as well as to managerial and leadership skills to build teams, integrate services, chair meetings, motivate team members and implement change. It is clear that the knowledge of pain management is not enough for the establishment and effective operation of the APS. Managerial knowledge and skills in team building, communication, group dynamics, change management are also an essential ingredient of achieving the paradigmatic change, the introduction of APS requires [27, 48]. Hungary can learn from this example to speed up the transformation process.

Financing difficulties are a general barrier to the improvement of patient care, and the APS is no exception. The adoption of the technology, its long term sustainability and development over time is an illusion without appropriate financing, equipment and human resources [49]. It is a disturbing finding of our study that neither of the identified APS receive extra financing. Relying only on the commitment and intrinsic motivation of health professionals might be enough to maintain APS in a couple of first mover hospitals, but unlikely to provide a solid background to a country-wide rollout of the initiative.

The operational costs of the APS are mainly personnel coming from regular visits and consultations by the bedside, but the costs of equipment and supplies, such as mobile ultrasound machines, local anesthetics, cannulas and reanimation devices, have to be covered, as well [47]. While low cost implementations involving only specialised pain nurses might be suitable for the provision of routine pain care, a limited competency APS not only lowers its operational costs, but also its effectiveness, accessibility and future development potential [33]. It would not be wise to base the investment decisions solely on the consideration of capital and actual running costs of the service without looking at the benefits, both in terms of health gain and potential future cost savings. The added value of APS, which comes from a range of health benefits and reduced resource use, such as the better pain control, lower opiate consumption, quicker return of intestinal functions, faster recovery, reduced length of hospital stay and in intensive therapy, decreased incidence of long term, chronic pain, as well as longer survival, especially among certain malignancies [47], well offset the upfront investment costs and running expenses [50].

Further, high quality pain relief is a strong determinant of overall patient satisfaction with the provided hospital care, so from the managerial perspective, it is an important source of good publicity for the organization concerned [47]. Nevertheless, the feasibility of an outcome based, pay for performance financing is limited by the lack of consensus on appropriate outcome indicators, which can be measured objectively and suitable for comparison [47].

In addition to the main obstacles discussed before, the respondents of our study also mentioned the low patient turnover, and small hospital size as a barrier to implementation. This factor is not unknown in the literature, either [21, 31].

Finally, it is encouraging that the majority of our respondents would be willing to take an active part in the establishment of APS in their hospitals. This enthusiasm is a good starting point to build on, however, it is hardly enough, in itself, to see a rapid adoption of the technology in Hungary. The availability of financial and human resources is a critical success factor, especially anesthesiologists, who are acknowledged as the owners and managers by most of the respondents of our study. This agrees well with the literature, since anesthesiologists are in the driving seat of APS virtually in every country, where hospitals already operate such services [21, 47].

## Limitations

The design and implementation of the study imply some limitations, which should be considered, when the findings of this research are interpreted. First, the survey was based on a self-completion questionnaire, which does not exclude the possibility that the respondents provided biased answers to put their hospitals, departments and units in a more favourable light. Given that the respondents reported on and evaluated of their own work, overestimation due to partiality might also biased the results. Second, the willingness to respond might have been limited by the overload of, or time pressure on hospital workers, and the relatively short data collection period. Nonetheless, taking into account the general shortage of health professionals in Hungary, the achieved response rate should be considered good. Third, the response rates could not be broken down to professions, because the National Healthcare Service Center, which was the supervisor of the government owned hospital under the Ministry of Human Capacities, managed the distribution and collection of questionnaires, and returned the completed questionnaires in one pile, without sorting them according to hospitals. Further, hospitals are allowed to operate their departments jointly or in a matrix structure in Hungary, so it is possible that hospitals have separate licences to operate different specialities, but in fact run them together in one organizational unit. Fourth, certain private hospitals possess the necessary licence to provide inpatient care, but in reality they only operate one-day surgery. These hospitals were obviously left out of the study, but reaching the relevant private hospitals proved to be the most challenging part of data collection. We were not very successful in motivating them to fill in the questionnaire.

We are convinced that despite these limitations our study provides a valuable insight into the status of postoperative pain management in Hungary, and has identified the challenges of a countrywide adoption of APS by hospitals. The findings of this research may orient the health policy on APS not just in Hungary, but also in other Central and Eastern European countries with similarly scarce financial resources. More affluent countries may benefit from the experiences of middle income countries, too, as the pressure coming from the limited financial space is a strong motivational factor to cost-effective innovation.

## Conclusions

In summary, the presence of APS in Hungarian hospitals is not at all widespread. The two teams identified in our sample by and large meet the criteria determined in the literature. Their wider adoption is not impossible, but has several obstacles, including the shortage of health professionals in the Hungarian health system, financing difficulties and educational deficiencies. Taking into account the Hungarian context, the Swedish model of Rawal would be the most feasible option, since it is based on nurses under the supervision of an anesthesiologist. This cost-effective version of the APS fits better to the realities of Central and Eastern European countries, with its lower cost and decent patient outcomes [37]. Further research is needed into the policy and practice of postoperative pain management to support the wider adoption of this organizational technology.

## Supporting information

**S1 Table. The characteristics of Hungarian APS teams.**
(XLSX)

**S2 Table. Answers deducted from filled questionnaires.**
(XLSX)

**S1 File. APS questionnaire in Hungarian.**
(DOCX)

**S2 File. APS questionnaire translated to English.**
(DOCX)

## Acknowledgments

We are very grateful for all the participating owners, health care providers, and individual health professionals, without whom this study would not have been possible. The authors owe special thanks to the National Healthcare Service Center (currently National Directorate General of Hospitals), which supported the data collection process among government hospitals under the Ministry of Human Capacities (currently under the Ministry of Interior).

## Author Contributions

**Conceptualization:** Orsolya Lovasi.

**Data curation:** Orsolya Lovasi.

**Formal analysis:** Orsolya Lovasi.

**Investigation:** Orsolya Lovasi, Judit Lám.

**Methodology:** Réka Schutzmann.

**Project administration:** Orsolya Lovasi.

**Supervision:** Judit Lám, Péter Gaál.

**Visualization:** Péter Gaál.

**Writing – original draft:** Orsolya Lovasi, Péter Gaál.

**Writing – review & editing:** Judit Lám, Réka Schutzmann, Péter Gaál.

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
