## [Decision Letter · Decision Letter 0]

15 Jun 2021

PONE-D-21-15759

Acute Pain Service in Hungarian Hospitals

PLOS ONE

Dear Dr. Lovasi,

Thank you for submitting your manuscript to PLOS ONE. After careful consideration, we feel that it has merit but does not fully meet PLOS ONE’s publication criteria as it currently stands. Therefore, we invite you to submit a revised version of the manuscript that addresses the points raised during the review process.

We look forward to receiving your revised manuscript.

Kind regards,

Gabor Erdoes, M.D., Ph.D.

Academic Editor

PLOS ONE

Journal Requirements:

Reviewers' comments:

Reviewer's Responses to Questions

**Comments to the Author**

1. Is the manuscript technically sound, and do the data support the conclusions?

Reviewer #1: Yes

2. Has the statistical analysis been performed appropriately and rigorously? 

Reviewer #1: I Don't Know

3. Have the authors made all data underlying the findings in their manuscript fully available?

Reviewer #1: Yes

4. Is the manuscript presented in an intelligible fashion and written in standard English?

Reviewer #1: Yes

5. Review Comments to the Author

Reviewer #1: Manuscript Number PONE-D-21-15759

Acute Pain Service in Hungarian Hospitals

The authors focus on an important issue: pain treatment can be more effectively managed by a multidisciplinary team of health professionals, referred to as Acute Pain Service (APS). The main objective of the present study was to evaluate the prevalence of APS and explore different factors (e.g., structure, barriers, conducive factors) in Hungarian hospitals.

To investigate these questions, a survey was conducted among all the 72 Hungarian hospital which had surgical inpatient care unit (with good response rate).

APS is much more accepted in developed countries, and I personally believe that it is important to start to raise awareness why it is so beneficial to work with APS teams in Hungary/ Central and Eastern European countries (it could increase the quality of postoperative pain treatment, decrease pain and the prevalence of side effects, reduces the length of stay etc.). The present paper did a good job in this regard, but I have some concerns about the Methods and Results sections. Since this is a research article, I would expect more results based on statistical analyses (but it is not clear how the data was analyzed). Clarifications by the authors would help improve the paper.

Detailed comments and/or suggested revisions for the authors:

Abstract:

- I would avoid using the word “irksome” – burdensome?

Introduction:

- It would be important to use the new/revised definition of pain (according to IAPS) in this paper. For a reference see:

https://www.iasp-pain.org/PublicationsNews/NewsDetail.aspx?ItemNumber=10475

doi: 10.1097/j.pain.0000000000001939

- Again, I would avoid using the word “irksome” – burdensome?

- Chronic pain should be defined as well (within parentheses)

- I would like to read more about how acute (and chronic) pain can affect the central nervous system and brain (structures and functions) as well. 1-2 sentences with reference

- “they are also an important factor “– important factors (plural)

- “Petti [7], the frequency of prolonged postoperative pain can be between 5% to 85%, depending on the type of the operation” –– Can you give some examples? Is there a specific type of operation which usually relates to prolonged postoperative pain?

- “whose number worldwide was estimated to exceed 300 million per annum in 2012” – Is there any more recent data/reference available?

- “decreases the pain scores of patients” sounds a bit informal – The authors should rephrase this (e.g., decreases the pain intensity in patients)

- The aim of our study: the authors should state here that the study was done using a survey (this should not be first mentioned in the Methods section)

Methods

- “We identified published questionnaires [1, 18, 19, 27-28], selected the relevant questions and translated and adapted these to the Hungarian context” – Were these questions translated back into English and compared to the original versions?

- “The collected data were processed and analyzed by SPSS 24.0.” – citation needed here, this is not the proper way to report what software was used for the analysis

- The tables should be better formatted –They occupy to much space relative to the information they demonstrate. Words should not be separated in a Table unless it is absolutely necessary

- The Methods section is missing a Data Analysis section. There is no information available about how the analysis was done, what methods were used exactly to analyze the questionnaires, and there is no indication if any or which assumptions were tested etc. It seems that the study is based mainly on descriptive statistics.

- Did the participants get any compensation for completing the survey? (if not this should be stated as well)

Results:

- “Among the respondent hospitals without established APS (N=115), the overwhelming majority (82.6%, n=95) think that such multidisciplinary teams would be necessary, while less than one fifth (17.4%, n=20) think that they are not needed” – What do we know about these 2 groups in terms of position and speciality?

- Pain scores should be defined throughout the paper (and under the Tables in the Notes section)

- Minimum and maximum values should be stated where applicable

Discussion:

- “From a cross-country comparative perspective, we can say that the wealthier European countries, with high level of public spending on health started the introduction of APS decades ago, and the technology has been gaining ground at a fast pace ever since.” – the authors need to use some references here. On what study is this statement based on?

- Limitation should be a separate section and then Conclusion should be the last section

The quality of Figure 1 needs to be improved

6. PLOS authors have the option to publish the peer review history of their article (what does this mean?). If published, this will include your full peer review and any attached files.

Reviewer #1: No

---

## [Author Response · Author response to Decision Letter 0]

29 Jul 2021

Thank you for your comments to our manuscript, according to which, we have prepared a revised version. We have addressed the editorial questions and requests as follows:

1. Please include additional information regarding the survey or questionnaire used in the study and ensure that you have provided sufficient details that others could replicate the analyses. For instance, if you developed a questionnaire as part of this study and it is not under a copyright more restrictive than CC-BY, please include a copy, in both the original language and English, as Supporting Information.

The complete questionnaire of the survey consisted of 67 questions. In the current manuscript only the group of questions referring to the APS teams have been processed. We have attached these 29 questions as supporting information files in Hungarian and English. 

The questionnaires were sent via e-mail to the owners of the hospitals. The attached cover letter contained the necessary information which was important for filling it out, and information about the purpose of the study and use of the data collected. We asked the owners to have the questionnaires completed per discipline and returned back to us. By returning the questionnaires they agreed to participate in our study. Filling in the questionnaire was anonymous and voluntary, we did not collect personal data, we dealt exclusively with the existence of a structure related to institutional pain relief, therefore there was no need to obtain a written informed consent from those, who completed the questionnaire. The ethical permission was acquired and we have indicated this in the manuscript. The ethical committee did not require us to obtain a written informed consent.

3. Upon re-submitting your revised manuscript, please upload your study’s minimal underlying data set as either Supporting Information files or to a stable, public repository and include the relevant URLs, DOIs, or accession numbers within your revised cover letter.

Responses from the two hospitals operating APS and exclusively the data used for this survey are attached as supplementary information in excel tables in the language of research. 

We also thank for the reviewer’s work and valuable comments. We report our responses to these as follows:

R1. Since this is a research article, I would expect more results based on statistical analyses (but it is not clear how the data was analyzed). 

This explorative survey was the first of this kind in Hungary, and, as APS is present only in a small number of Hungarian hospitals, it was not possible to carry out a comparative study with more profound statistical analysis. The purpose of this survey was to explore a starting situation, for which descriptive statistical methods are appropriate.

R2. Abstract: I would avoid using the word “irksome” – burdensome?

We have changed the text accordingly.

R3. Introduction: It would be important to use the new/revised definition of pain (according to IAPS) in this paper. For a reference see:

https://www.iasp-pain.org/PublicationsNews/NewsDetail.aspx?ItemNumber=10475

doi: 10.1097/j.pain.0000000000001939

R4. Again, I would avoid using the word “irksome” – burdensome? 

We have changed the text.

R5. Chronic pain should be defined as well (within parentheses)

We have complemented the text.

R6. I would like to read more about how acute (and chronic) pain can affect the central nervous system and brain (structures and functions) as well. 1-2 sentences with reference

We have added some information with reference to the text as requested.

R7. “they are also an important factor “– important factors (plural)

We have corrected the text.

R8. “Petti [7], the frequency of prolonged postoperative pain can be between 5% to 85%, depending on the type of the operation” –– Can you give some examples? Is there a specific type of operation which usually relates to prolonged postoperative pain?

We have complemented the text.

R9. “whose number worldwide was estimated to exceed 300 million per annum in 2012” – Is there any more recent data/reference available?

We have looked for it again, but have not found more recent data. According to the database of EUROSTAT, the most recent data are from 2016, which are not complete due to the deficiency of data service. In the database of WHO we have not found any data regarding the number of operations.

R10. “decreases the pain scores of patients” sounds a bit informal – The authors should rephrase this (e.g., decreases the pain intensity in patients)

We have corrected the text.

R11. The aim of our study: the authors should state here that the study was done using a survey (this should not be first mentioned in the Methods section)

We have inserted the required sentence in the introductory section.

R12. Methods: “We identified published questionnaires [1, 18, 19, 27-28], selected the relevant questions and translated and adapted these to the Hungarian context” – Were these questions translated back into English and compared to the original versions?

We did not consider it necessary to translate the questions back into the original language as the language of filling out was Hungarian and the questions were adapted in a way to be interpretable in the Hungarian Health Care System. Furthermore, we did not want to carry out international comparison.

R13. “The collected data were processed and analyzed by SPSS 24.0.” – citation needed here, this is not the proper way to report what software was used for the analysis.

We have complemented the text.

R14. The tables should be better formatted –They occupy to much space relative to the information they demonstrate. Words should not be separated in a Table unless it is absolutely necessary

The tables have been changed, as requested. 

R15. The Methods section is missing a Data Analysis section. There is no information available about how the analysis was done, what methods were used exactly to analyze the questionnaires, and there is no indication if any or which assumptions were tested etc. It seems that the study is based mainly on descriptive statistics.

As we have pointed out earlier, this explorative survey was the first of this kind in Hungary, and, as APS is present only in a small number of Hungarian hospitals, it was not possible to carry out a comparative study with more profound statistical analysis. The purpose of this survey was to explore a starting situation, for which descriptive statistical methods are appropriate.

R16. Did the participants get any compensation for completing the survey? (if not this should be stated as well)

The responders did not get any compensation. We have specified this in the manuscript, as requested.

R17. Results: “Among the respondent hospitals without established APS (N=115), the overwhelming majority (82.6%, n=95) think that such multidisciplinary teams would be necessary, while less than one fifth (17.4%, n=20) think that they are not needed” – What do we know about these 2 groups in terms of position and speciality?

We have complemented the text, as requested.

R18. Pain scores should be defined throughout the paper (and under the Tables in the Notes section)

R19. Minimum and maximum values should be stated where applicable

We have complemented the text accordingly.

R20. Discussion: “From a cross-country comparative perspective, we can say that the wealthier European countries, with high level of public spending on health started the introduction of APS decades ago, and the technology has been gaining ground at a fast pace ever since.” – the authors need to use some references here. On what study is this statement based on?

We have added the requested citations.

R21. Limitation should be a separate section and then Conclusion should be the last section

We have changed the text accordingly.

R22. The quality of Figure 1 needs to be improved

We have used PACE before submitting the manuscript to check the quality of our figure as suggested by the journal. Additionally, in the revised version we have adjusted the text to improve clarity. If any further correction is needed, please provide us more guidance on what to change.

---

## [Decision Letter · Decision Letter 1]

27 Aug 2021

PONE-D-21-15759R1

Acute Pain Service in Hungarian Hospitals

PLOS ONE

Dear Dr. Lovasi,

Thank you for submitting your manuscript to PLOS ONE. After careful consideration, we feel that it has merit but does not fully meet PLOS ONE’s publication criteria as it currently stands. Therefore, we invite you to submit a revised version of the manuscript that addresses the points raised during the review process.

We look forward to receiving your revised manuscript.

Kind regards,

Gabor Erdoes, M.D., Ph.D.

Academic Editor

PLOS ONE

Journal Requirements:

Reviewers' comments:

Reviewer's Responses to Questions

**Comments to the Author**

1. If the authors have adequately addressed your comments raised in a previous round of review and you feel that this manuscript is now acceptable for publication, you may indicate that here to bypass the “Comments to the Author” section, enter your conflict of interest statement in the “Confidential to Editor” section, and submit your "Accept" recommendation.

Reviewer #1: All comments have been addressed

Reviewer #2: All comments have been addressed

2. Is the manuscript technically sound, and do the data support the conclusions?

Reviewer #1: Yes

Reviewer #2: Yes

3. Has the statistical analysis been performed appropriately and rigorously? 

Reviewer #1: Yes

Reviewer #2: Yes

4. Have the authors made all data underlying the findings in their manuscript fully available?

Reviewer #1: (No Response)

Reviewer #2: No

5. Is the manuscript presented in an intelligible fashion and written in standard English?

Reviewer #1: (No Response)

Reviewer #2: Yes

6. Review Comments to the Author

Reviewer #1: I have accepted the revision.

However, for future submissions, the authors should highlight the changes they make in the revised text (it is very difficult for the reviewers to find all the newly added information in the text).

Also, when the authors write their responses, they should list where they made the corrections (on what page), moreover, they should show these changes even in the "Response to the reviewers' comments" document.

All in all, the revision process needs to be improved by the authors.

Reviewer #2: The authors wrote an article about use of APS services using a survey among all the 72 Hungarian hospitals, which had surgical inpatient care.

Although I am not the peer reviewer that made the initial comments, I carefully read the article, the comments and the responses/changes. I think that the response of the review comments was adequate.

I think this is an interesting article and technically sound. As mentioned in limitations, only descriptive statistics were possible, because the Ministry of Human Capacities, managed the distribution and collection of questionnaires, and returned the completed questionnaires in one pile, without sorting them according to hospitals and only two hospitals had APS teams. However, I miss information about the characteristics of hospitals that did not respond? All private hospitals? And if information about respondents (of different wards in different hospitals) is given, please do not suggest that this is from hospitals (for example in page 13, line 252).

Furthermore, the introduction is quite long, about 3,5 pages. I think this could be more concisely written.

7. PLOS authors have the option to publish the peer review history of their article (what does this mean?). If published, this will include your full peer review and any attached files.

Reviewer #1: No

Reviewer #2: No

---

## [Author Response · Author response to Decision Letter 1]

1 Sep 2021

Reviewer #1

However, for future submissions, the authors should highlight the changes they make in the revised text (it is very difficult for the reviewers to find all the newly added information in the text).

Also, when the authors write their responses, they should list where they made the corrections (on what page), moreover, they should show these changes even in the "Response to the reviewers' comments" document.

All in all, the revision process needs to be improved by the authors.

We are sorry, that our responses were difficult to follow, we will work to improve our efficiency and clarity during future revisions. However, a marked-up copy of the manuscript that highlights changes made is a mandatory requirement during revision, and we prepared it accordingly, and attached to our submission, so we thought this was enough to highlight the changes. It would have been no problem for us to indicate the relevant page numbers, and include the actual changes in the “response document”. We were not aware that this was a requirement and happy to do so in the future, as we have done in this current document.

Reviewer #2

The authors wrote an article about use of APS services using a survey among all the 72 Hungarian hospitals, which had surgical inpatient care.

Although I am not the peer reviewer that made the initial comments, I carefully read the article, the comments and the responses/changes. I think that the response of the review comments was adequate.

I think this is an interesting article and technically sound. As mentioned in limitations, only descriptive statistics were possible, because the Ministry of Human Capacities, managed the distribution and collection of questionnaires, and returned the completed questionnaires in one pile, without sorting them according to hospitals and only two hospitals had APS teams. However, I miss information about the characteristics of hospitals that did not respond? All private hospitals? And if information about respondents (of different wards in different hospitals) is given, please do not suggest that this is from hospitals (for example in page 13, line 252).

Although, we did not detail it in the text, in Table 2 we added information about the Hungarian hospitals (included in the survey and responding). In total, 20 hospitals did not return filled questionnaires, 12 government hospitals, two university clinics, one specialized hospital (for professional athletes) and five private for-profit hospitals. This sentence was also added to the text, in lines 184-185.

Additionally, we have checked the text throughout regarding hospitals vs. departments/wards, and improved the clarity of the text as “Among the respondent departments from hospitals without established APS (N=115),” in line 252. We have not conquered this problem elsewhere in the text.

Furthermore, the introduction is quite long, about 3,5 pages. I think this could be more concisely written.

We have shortened the introduction as requested. However, we feel that the information included in the introduction is necessary to understand the topic for all readers who are interested but not specialized in pain management. Some of the additional content of the introduction was incorporated as a response to the request of reviewer 1.

The following sentences were deleted:

“Because pain causes suffering, it is a priority symptom among patients and its relief is an important determinant of patient satisfaction, so much so that professional organizations, such as the American Pain Society, labelled it the „fifth vital sign” in addition to body temperature, pulse rate, respiration rate and blood pressure [4-6].” (from lines 49-53)

“Prolonged pain can cause disability, whose costs are also high. In the USA, for instance, it was estimated to be 34 000 USD per patient per annum [1]”. (from lines 63-65)

“reducing length of stay, as well as to improve the quality of pain therapies through innovation” (from line 95)

---

## [Editor Report · Decision Letter 2]

6 Sep 2021

Acute Pain Service in Hungarian Hospitals

PONE-D-21-15759R2

Dear Dr. Lovasi,

We’re pleased to inform you that your manuscript has been judged scientifically suitable for publication and will be formally accepted for publication once it meets all outstanding technical requirements.

Kind regards,

Gabor Erdoes, M.D., Ph.D.

Academic Editor

PLOS ONE

---

## [Editor Report · Acceptance letter]

14 Sep 2021

PONE-D-21-15759R2 

Acute Pain Service in Hungarian Hospitals 

Dear Dr. Lovasi:

I'm pleased to inform you that your manuscript has been deemed suitable for publication in PLOS ONE. Congratulations! Your manuscript is now with our production department. 

Kind regards, 

on behalf of

Prof. Dr. Dr. Gabor Erdoes 

Academic Editor

PLOS ONE